# A Mediation Analysis to Identify Links between Gut Bacteria and Memory in Context of Human Milk Oligosaccharides

**DOI:** 10.3390/microorganisms9040846

**Published:** 2021-04-15

**Authors:** Stephen A. Fleming, Jonas Hauser, Jian Yan, Sharon M. Donovan, Mei Wang, Ryan N. Dilger

**Affiliations:** 1Traverse Science, Inc., Champaign, IL 61820, USA; rdilger2@illinois.edu; 2Société des Produits Nestlé SA, 1000 Lausanne, Switzerland; jonas.hauser@rdls.nestle.com; 3Nestlé Product Technology Center Nutrition, CH-1800 Vevey, Switzerland; jy435@cornell.edu; 4Department of Food Science and Human Nutrition, University of Illinois, Urbana, IL 61801, USA; sdonovan@illinois.edu (S.M.D.); meiwang@illinois.edu (M.W.); 5Division of Nutritional Sciences, University of Illinois, Urbana, IL 61801, USA; 6Piglet Nutrition and Cognition Laboratory, Department of Animal Sciences, University of Illinois, Urbana, IL 61801, USA; 7Neuroscience Program, University of Illinois, Urbana, IL 61801, USA

**Keywords:** microbiome, cognition, oligosaccharide, prebiotic, neurotransmission, glutamate, GABA, memory, bacteria, behavior

## Abstract

Elucidating relationships between the gut and brain is of intense research focus. Multiple studies have demonstrated that modulation of the intestinal environment via prebiotics or probiotics can induce cognitively beneficial effects, such as improved memory or reduced anxiety. However, the mechanisms by which either act remain largely unknown. We previously demonstrated that different types of oligosaccharides affected short- and long-term memory in distinct ways. Given that the oligosaccharide content of human milk is highly variable, and that formula-fed infants typically do not consume similar amounts or types of oligosaccharides, their potential effects on brain development warrant investigation. Herein, a mediation analysis was performed on existing datasets, including relative abundance of bacterial genera, gene expression, brain volume, and cognition in young pigs. Analyses revealed that numerous bacterial genera in both the colon and feces were related to short- and/or long-term memory. Relationships between genera and memory appeared to differ between diets. Mediating variables frequently included GABAergic and glutamatergic hippocampal gene expression. Other mediating variables included genes related to myelination, transcription factors, brain volume, and exploratory behavior. Overall, this analysis identified multiple pathways between the gut and brain, with a focus on genes related to excitatory/inhibitory neurotransmission.

## 1. Introduction

During early life, neurodevelopment is strongly impacted by nutrition [1,2,3,4,5], among other factors. Of the components in human milk, oligosaccharides provide little to no direct nutritional value to the infant yet are the third most concentrated solid in human milk [6]. Human milk contains a greater variety and higher concentration of oligosaccharides than other mammals [7] and their concentrations are highly variable between mothers [8]. Their functions are multifaceted, impacting immunity [9], mucosal physiology [10], gut bacteria [11], and potentially brain development [12,13,14,15].

Regarding the impact on neurodevelopment, oligosaccharides of multiple origins have shown the capacity to alter human and animal behavior. Rats provided 2′-fucosyllactose (2′-FL) demonstrated vagal-dependent improved learning and memory [16,17], mice consuming chitosan-derived oligosaccharides showed attenuations in neurodegenerative-associated phenotypes [18], and a recent paper associated human milk levels of 2′-FL at 1 month with cognitive function at 24 months in human infants [19]. The potential mechanisms linking the gut and brain are myriad, and include alteration of the central neurotransmitter systems, alteration of the microbiota or microbial-derived metabolites, and direct vagal mediation. For example, the hippocampal serotonergic system is altered in germ-free animals and partially restored by colonization [20], and fecal microbial transplants (FMT) from schizophrenic patients to germ-free mice altered gamma-aminobutyric acid (GABA)-related metabolism in the brain and behavior [21]. Mice that demonstrated depressive-like behavior after a FMT from a non-obese diabetic donor mouse had increased fecal cresol, which has been shown to impair myelination in culture [22]. Short-chain fatty acids from microbial fermentation have been linked to anxiety and aggression [23,24], or conversely proposed as potential therapeutics [25]. Lastly, the vagus nerve mediates improvements to behavior after ingestion of oligosaccharides or probiotics [17,26], as vagotomy prevents pre-/probiotic-related improvements to behavior.

Desire to modulate the microbiota-gut-brain axis has pervaded numerous psychological disciplines: cognitive development [27], depression [28], autism [29,30], schizophrenia [21], and others. The development of the microbiome and its relationship to neurodevelopment is of special interest as the neonatal period represents a critical window when the both the microbiome and brain are sensitive to both distress and eustress, potentially shaping long-term development [31].

We previously demonstrated that consumption of human and non-human oligosaccharides significantly improved performance in a test of recognition memory in pigs [32,33], but such a benefit was specific to the type of oligosaccharide ingested and the type of memory tested (i.e., short or long term). Furthermore, oligosaccharide intake altered average brain volume and hippocampal gene expression [32,33]. Surprisingly, minimal differences between diets in relative abundance of bacterial genera in the colon were found [34]. We identified potential mechanistic relationships between neurotransmitter-related genes and recognition memory; however, these relationships were not consistently found in pigs fed different oligosaccharides. The objective of this study was to perform a secondary analysis on the same datasets to determine whether a relationship exists between the bacteria present in the ascending colon and feces and recognition memory, and, if so, whether that relationship was mediated by gene expression or magnetic resonance imaging outcomes in the context of oligosaccharide intake.

## 2. Materials and Methods

Procedures have been reviewed in detail in [32,33,34] and are summarized below.

### 2.1. Animals and Housing

All animal care and experimental procedures were in accordance with the National Research Council Guide for Care and Use of Laboratory Animals and approved by the University of Illinois at Urbana-Champaign Institutional Animal Care and Use Committee. Seventy-two intact male pigs (Camborough^®^ breed, Line 2 and Line 3) were artificially-reared from postnatal day (PND) 2 until PND 33 across six independent cohorts (n = 12 per cohort, each cohort separated in time). Pigs were randomized to each group such that litter representation and initial bodyweight (BW) were counterbalanced within each cohort and between each dietary group, respectively. All pigs were housed in master caging units that contained six individual stainless-steel cages (L × W × H of 87.6 × 88.9 × 50.8 cm) with clear, polycarbonate facades on three sides of the cage and vinyl-coated, expanded-metal flooring (Tenderfoot^®^, Minneapolis, MN, USA). The flooring allowed fecal and urinary output to drop to a collection system below the pan, thereby reducing the opportunity for fecal contamination of the diets or coprophagy. Pigs were able to see, smell, hear, and minimally touch one another. A towel and toy were included in each cage to provide enrichment, and all pigs were removed from cages and allowed to socialize with each other for approximately 30 min each day. From subjective observation, minimal transfer of fecal material between groups occurred during socialization.

All pigs were reared in the same room with ambient temperature maintained between 27 and 29 °C and a 12 h light/dark cycle maintained from 600 to 1800 h. Prior to placement in the artificial rearing system, pigs were administered 5.0 mL of *Clostridium perfingens* antitoxin C + D per the manufacturer’s recommendations (Colorado Serum Company, Denver, CO, USA), a standard veterinary procedure to prevent enterotoxemia [35]. At study conclusion (PND 33), pigs were anesthetized using a telazol:ketamine:xylazine solution (50.0 mg tiletamine plus 50.0 mg of zolazepam reconstituted with 2.50 mL ketamine [100 g/L] and 2.50 mL xylazine [100 g/L]; Fort Dodge Animal Health) by intramuscular injection at 0.03 mL/kg BW. After anesthetic induction, pigs were euthanized via intracardiac administration of sodium pentobarbital (86.0 mg/kg of body weight; Euthasol, Virbac Animal Health, Fort Worth, TX, USA). Two pigs (from the BMOS + HMO group) were removed from study due to failure-to-thrive (i.e., exhibited very low growth).

### 2.2. Dietary Treatments

Pigs (n = 12 per diet) were provided milk replacers reconstituted at 200 g of dry powder per 800 g of water. Reconstituted diets were analyzed to contain approximately 0 g/L oligosaccharide (OS) (control [CON], ProNurse^®^ Specialty Milk Replacer, Purina Animal Nutrition, Gray Summit, MO, USA), 5.79 g/L bovine milk oligosaccharides (BMOS, Nestlé Product & Technology center, Konolfingen, Switzerland), 1.23 g/L HMO (HMO, 0.81 g/L of 2′-fucosyllactose [2′-FL] + 0.42 g/L of Lacto-N-neotetraose [LNnT], Glycom, Hørsholm, Denmark), both bovine and human milk oligosaccharides (BMOS + HMO; 5.75 g/L of BMOS + 1.00 g/L of 2′-FL + 0.53 g/L of LNnT), 3.62 g/L oligofructose (OF, Orafti^®^ P95, Beneo-Orafti, Tiene, Belgium), or 3.41 g/L OF + 1.12 g/L 2′-FL (OF + 2′-FL). Briefly, the BMOS product was derived from bovine whey and composed primarily of galactooligosaccharide and trace amounts of 3′- and 6′-sialyllactose. All diets were formulated with the addition of lactose, such that each diet contained the same amount of total carbohydrate (See Table 1). The nutritional composition of the base formula and oligosaccharide content have been previously reported [32,33]. Though concentrations of each oligosaccharide differed, confounding the ability to differentiate between dose and type of oligosaccharide, the concentrations were chosen to remain consistent with previously conducted clinical trials on BMOS [36,37,38], 2′-FL [39,40], and OF [41,42]. Ultimately, the difference in lactose between diets minimally contributed to the metabolizable energy and growth was equivalent between groups [32,33].

Pigs received approximately 500 mL of experimental diets on the day of arrival to the rearing facility and were fed at a rate of 285 mL and 325 mL of reconstituted milk replacer per kg BW from PND 3-6 and PND 7-33, respectively. Bodyweight was recorded daily to accurately dose meals, which were administered 10 times per day, approximately every 100 min, between 1000 h and 0400 h using an automated feeding system. Feed refusals were not quantified. All pigs were allowed *ad libitum* access to water at all times.

### 2.3. Behavior

Pigs were tested on the novel object recognition (NOR) task using two different delays to assess short- and long-term recognition memory. Methods used were adapted from previous studies using this task in pigs from other labs [43,44,45,46], our own lab [47,48,49], and are described from this study [32,33]. Testing consisted of a habituation phase, a sample phase, and a test phase. During the habituation phase, each pig was placed in an empty testing arena for 10 min each day for two days leading up to the sample phase. In the sample phase, the pig was placed in the arena containing two identical objects and given 5 min for exploration. After a delay of 1 or 48 h (representing a short or long delay, respectively), the pig was returned to the arena for the test phase of the NOR task. During the test phase, the pig was placed in the arena containing one object from the sample phase and a novel object and allowed to explore for 5 min. Habituation trials began at PND 22 and testing on the sample phase began on PND 24. Recognition index, or the proportion of time spent with the novel object compared to total exploration of both objects, was used to measure recognition memory.

### 2.4. Magnetic Resonance Imaging (MRI)

All pigs underwent MRI procedures at PND 32 at the Beckman Institute for Advanced Science and Technology Biomedical Imaging Center using a Siemens MAGNETOM Trio 3T equipment with a Siemens 32-channel head coil. Methods described here are adapted from previous studies using MRI in pigs [50,51,52] and are published [32,33]. Each pig underwent imaging protocols only once. The pig neuroimaging protocol included three magnetization prepared rapid gradient-echo (MPRAGE) sequences and diffusion tensor imaging (DTI) to assess brain macrostructure and microstructure, respectively, as well as magnetic resonance spectroscopy (MRS) to obtain brain metabolite concentrations. In preparation for MRI procedures, anesthesia was induced using an intramuscular injection of telazol (50.0 mg of tiletamine plus 50.0 mg of zolazepam reconstituted with 5.0 DI water; Zoetis, Florham Park, NJ, USA) administered at 0.07 mL/kg BW, and maintained with inhalation of isoflurane (98% O_2_, 2% isoflurane). Pigs were immobilized during all MRI procedures. Visual observation of each pig’s well-being, as well as observations of heart rate, PO_2_ and percent of isoflurane were recorded every 5 min during the procedure and every 10 min post-procedure until animals recovered. Total scan time for each pig was approximately 60 min.

### 2.5. Hippocampal Gene Expression

As previously described [32,33], relative mRNA copy numbers on 93 genes in the hippocampus were quantified using the NanoString nCounter™ system (NanoString Technologies Inc., Seattle, WA, USA) according to the manufacturer’s instructions using 100 ng of RNA as the starting material. Using nSolver software (Version 4.0, NanoString Technologies Inc., Seattle, WA, USA), background subtraction using the median of all eight negative controls was followed by positive control normalization using the geometric mean of six positive controls and housekeeping normalization using the geometric mean of six housekeeping genes (Ribosomal Protein L19,RPL-19; Ribosomal Protein S18, RPS18; Actin β, ACTB; Tubulin β Class 1, TUBB; Glyceraldehyde-3-Phosphate Dehydrogenase, GAPDH; β-glucoronidase, GUS).

### 2.6. DNA Extraction

DNA was extracted from ascending colon (AC, henceforth interchangeably referred to as colonic) contents and feces by the QIAamp Fast DNA Stool Mini Kit (Qiagen, Valencia, CA, USA) in combination with bead beating on the FastPrep-24 System (MP Biomedicals, Carlsbad, CA, USA) as previously described [53]. DNA concentration was measured with a NanoDrop 1000 spectrophotometer (NanoDrop Technologies, Wilmington, DE, USA).

### 2.7. PCR Amplification and Sequencing of 16S rRNA Genes

PCR amplification and sequencing of 16S rRNA genes were performed at the DNA Services Lab at the University of Illinois at Urbana-Champaign according to methods previously described [54]. Briefly, the V3-V4 region of bacterial 16S rRNA genes were amplified using primers V3f (5′-CCTACGGGAGGCAGCAG-3′) and V4r (5′-GGACTACHVGGGTWTCTAAT-3′) as previously described. The amplicons were mixed in equimolar concentrations and sequenced on an Illumina MiSeq flow cell for 251 cycles from each end of the fragments using a MiSeq 500-cycle sequencing kit version 2 (2 × 250 nt paired end reads).

### 2.8. Sequence Processing

Sequences were demultiplexed at the sequencing facility with the bcl2fastq v2.17.1.14 Conversion Software (Illumina, San Diego, CA, USA), and allowed 0 mismatches in the barcode sequences. De-multiplexed forward (read 1) and reverse reads (read 2) were processed using the QIIME software package as described previously [54,55]. Briefly, the paired-end reads were merged, quality filtered, and split into libraries at Phred quality score ≥ 25. Operational taxonomic unit (OTU) assignment, representative sequence picking, and chimera removal were performed as described by Monaco et al., [54]. An OTU table was created and further filtered to remove non-aligned and chimeric OTUs and singletons [54]. The representative sequence of each OTU was assigned to different taxonomic levels based on Greengenes taxonomy (gg_13_8) with the use of the Ribosomal Database Project naïve Bayesian rRNA Classifier at 80% confidence level [56].

### 2.9. Statistical Analysis

#### 2.9.1. Data Cleaning and Variable Selection

Genes expressed below threshold (defined as the mean 2 standard deviations below the negative controls) were omitted from analysis. Microbial genera with a median relative abundance of less than 0.05% were omitted from analysis on a group-wise basis (i.e., if genera median abundance was above 0.05% in a dietary group, it was retained for that group only). Given extremely high multicollinearity, DTI measures of axial, medial, and radial diffusivity were omitted from the analysis, with only fractional anisotropy retained as the primary DTI outcome. All variables analyzed are shown in Appendix A. Missing values were imputed with the median of that group. Values of zero or observations that were not quantified (e.g., non-compliance during behavioral testing) were not considered as “missing” and were not imputed. Observations that were not quantified were not included in the analysis. After cleaning, a total of 70 pigs and 289 variables remained. The dataset contained 19,186 observations, of which 46 were not quantified (0.24%), and 998 were missing (5.2%). For a breakdown of sample size and missingness per group and variable, see Appendix A.

#### 2.9.2. Mediation Analysis

All data cleaning and statistics were conducting using R software [57]. Ordinary Least Squares regression coefficients were constructed using the stats package. Mediation analyses used the R package “Mediation” [58], with a bootstrap sample size of 2000 and 95% confidence interval estimates constructed using the percentile method. Predictor variables included microbial genera from both the ascending colon (62 variables) and fecal samples (63 variables). Here, the same genera from a different sampling location were included as separate variables. Mediating variables included gene expression (73 variables), MRI (60 variables), and behavioral variables (28 variables) of exploration during the novel object recognition task. The predicted variable was the recognition index from either a short- or a long-delay. All mediations analyses were performed on each diet group by using a sub-group analysis. All variables were rank transformed prior to mediation. A depiction of the model and equations are shown in Figure 1. All heatmap visualizations were created using the package “ComplexHeatmap” [59].

## 3. Results

A descriptive analysis of bacterial composition at the phylum and genus level is provided in Figure 2. Given this is a secondary analysis, reports of the impact of the diet on 16S sequencing outcomes, gene expression, MRI, and behavior are reported elsewhere [32,33,34]. Due to the volume of data generated, only mediations where the confidence interval of the indirect effect did not include zero are discussed.

### 3.1. Short-Term Memory

No variables were identified as mediators within the CON or BMOS groups. Mediating variables in the HMO group were dominated by two types: GABA-related (*GAD*, glutamate decarboxylase; *SLC32A1*, vesicular GABA transporter [*VGAT*]) and cortical volume (Figure 3). With the exception of ascending colon *Lactobacillus* relative abundance, all genera were from fecal samples. Mediating variables in the BMOS + HMO group were mostly volume of the pons or novel object visit time, which were related to the relative abundance of *Prevotella*, *Lactobacillus*, and *Desulfovibrio* in the ascending colon. In the OF group, whole brain volume mediated the relationship between fecal *Prevotella* and short-term memory.

In pigs fed OF + 2′-FL, mediators included several different neurotransmitter-related genes (*5HTR2*, Serotonin Receptor 2A; *CHRNA7*, Cholinergic Receptor Nicotinic Alpha 7 Subunit; *GABRA5*, GABA receptor Subunit Alpha-5; *GRIA2*, Glutamate Ionotropic Receptor AMPA Type Subunit 2; *GRIN2D*, *NMDAR2D*), and mediated relationships between colonic *Prevotella*, *Anaerotruncus*, *Megasphaera*, and short-term recognition memory.

### 3.2. Long-Term Memory

No mediating variables between bacterial genera and long-term memory were identified in the CON and OF + 2′-FL groups (Figure 4). In the other groups, a few variables tended to mediate multiple pathways. In the BMOS group, myelin-related genes (*MAG*, myelin-associated glycoprotein; *PLP*, proteolipid protein; *MBP*, myelin basic protein) mediated the relationship between fecal *Ruminococcus*, *Escherichia,* colonic *Butyricicoccus,* and long-term memory. Transcription-related genes (*CREBBP*, cyclic AMP response element binding protein [CREB] binding protein; *NR4A2*, nuclear receptor-related 1 protein [*NURR1*]) acted as mediators between long-term memory and colonic *Alistipes,* and fecal *Ruminococcus, Synergistes*, *Blautia*, unclassified *Veillonellaceae*, and *Megasphaera*.

In the HMO group, total distance moved in the novel object recognition task acted as a mediator for more predictors than any other variable. Predictor variables tended to be colonic genera (eight variables), rather than fecal genera (three variables). Additionally, *GABRD* (GABA type A receptor delta subunit) and relative volume of the corpus callosum mediated the relationship between colonic *Eubacterium* and fecal unclassified *Paraprevotellaceae*.

In pigs fed BMOS + HMO, mediating variables were dominated by GABA/Glu-related genes (*GABRA2*, GABA type A receptor alpha 2 subunit; *GRIN2B*, *NMDARD2B/GluN2B*; *SLC17A6*, vesicular glutamate transporter 2 [*VGLUT2*]; *SLC17A8*, vesicular glutamate transporter 3 [*VGLUT3*]), or mean novel object visit time. In the OF group, absolute volume of the caudate mediated the relationship between colonic *Bilophila* and long-term memory.

### 3.3. Suppressor Variables

Numerous mediating variables acted as suppressor variables, as their inclusion in the model increased the strength of the relationship between host bacteria and memory (Appendix A). These variables themselves were weakly associated with memory. They included measures of exploratory behavior (mean visit time, total distance moved), gene expression (mostly related to GABA/Glu or myelination), fractional anisotropy, magnetic resonance spectroscopy, or brain volume. These variables were present across a range of dietary groups for both short- and long-term memory, and nearly half of them were related to MRI outcomes.

## 4. Discussion

The goal of the present analysis was to uncover the complexity of the relationships between the previously observed findings linking dietary oligosaccharides and neurocognitive development [32,33], with a focus on gut bacteria. The relationships between each of these variables were not clear, however. For example, the human milk oligosaccharides 2′-FL and LNnT improved short-term, but not long-term memory, and likewise a combination of human and bovine milk oligosaccharides improved long-term, but not short-term memory [33]. Furthermore, the effects of these oligosaccharides on brain volume did not follow an easily discernible pattern. However, we did find that the relationship between gene expression and memory was dependent on the type of oligosaccharide consumed. Given that oligosaccharides have been shown to impact the composition and/or metabolic activity of the gut microbiota, often simultaneously with an improvement in behavioral outcomes, we hypothesized that gut bacteria may impact cognition by acting through a mediating variable.

Here, we combined datasets containing relative abundance of colonic/fecal bacteria, brain volume and chemistry, gene expression, and exploratory behavior using a mediation analysis to identify potential paths between the host microbiota and cognition. At a high level, we found that mediating variables varied with oligosaccharide intake, but tended to appear in similar patterns. Importantly, even though the median relative abundance of genera were similar between groups, the relationships these genera exhibited with mediating and predictor variables appeared to differ between groups.

Prior to discussing the biological importance of (or lack thereof) these mediations, several limitations to the present analysis should be addressed. Given the presence of high skew and outliers in several of the datasets, especially 16S sequencing data, we chose to rank transform all variables prior to regression. Hauke and Kossowski provide a helpful explanation on the consequences of rank transformation in context of Pearson and Spearman correlation [60]. While the use of Spearman correlation (and by extension linear regression on ranked variables) can lessen the impact of outliers and skewness while simultaneously allowing detection of monotonic relationships between 2 variables, it does have several limitations. Primarily, loss of information regarding variability and the spacing between variables: the distance between each observation is equal in a ranked variable, whereas in reality the distance between observations may be extremely variable. Furthermore, the data become less reproducible. In other words, future research will be unable to replicate our exact findings as the rank of a specific observation is only relevant within context of the raw data collected in this study. However, we felt the benefit of being able to identify both linear and non-linear relationships, reduced weighting of outliers, and reduced impact on skewness outweighed these drawbacks.

Herein, we used a sub-group analysis as an alternative to moderated mediation to identify the impact of a multicategorical variable (oligosaccharide intake across 6 different diets) on a mediation. Although a multicategorical, moderated mediation would be preferable to model, not all variables were equally present between groups (e.g., some microbial genera were completely absent in one group and abundant in another). Furthermore, attempts to create a conditional process model [61] were precluded by the inadvertent creation of rank-deficient matrices or singular inverse matrices. Though not clearly identified, these issues may have arisen due to high multicollinearity and interactions between predictor variables in the model that centering was unable to overcome. To avoid these issues a sub-group analysis was used, which presents its own limitations. As described by Hayes [62], such a method should not be used when: (1) the hypothesis is specific to moderation of a specified path; (2) a causal steps approach is used (i.e., mediation is only estimated after a relationship between X and Y, X and M, or M and Y are established); (3) sample size between groups differs; and (4) the moderating variable is not categorical. Here, we did not state an *a priori* hypothesis regarding which path would be moderated, did not perform a causal steps analysis, and the moderator chosen (diet) was categorical. Thus, we violated the third criterion, as not all groups contained equal sample size, creating an inequality in power to detect a mediation between groups. Therefore, the presence or absence of a mediation in one group compared to another should not be interpreted definitively that the mediation was moderated at the group level.

We chose not to interpret mediations as “partial” or “full”, due in part to the inequality in power between groups, as smaller sample sizes are more likely to result in the detection of full rather than partial mediations, artificially inflating the probability that a full mediation would be observed. Furthermore, despite conducting numerous mediations, we chose not to use a false discovery rate correction. Our reasoning was as follows: (1) small sample size resulted in lower power, increasing the probability of detecting false negatives (i.e., high likelihood that true mediating variables were undetected); and (2) adjusting for multiple comparisons would further increase the probability of detecting false negatives. Rather, our opinion is that in a high-level exploratory analysis, significant results should be interpreted in context of the literature and followed by replication and greater statistical power to confirm whether the observed effects are likely to have been false. With these limitations in mind the results should be interpreted in context of what can reasonably (i.e., theoretically and practically) be inferred from the present analysis.

### 4.1. Evidence of Common Paths to Cognition

#### 4.1.1. GABA and Glutamatergic Mediators

Perhaps one of the more surprising findings was the convergence of multiple bacterial genera on a few classes of mediators. For example, in the HMO group, the relationship between bacterial genera and short-term memory was mediated by two GABAergic genes (*GAD* and *SLC32A1*) and both the absolute and relative volumes of the left and right cortices (Figure 3). Genera (mostly fecal, some colonic) such as *Lactobacillus*, *Oscillospira*, and unclassified *Ruminococcaceae*, *Christensenellaceae*, and *Elusimicrobiaceae* converged onto a combination of these two types of mediators. Furthermore, the direction of the relationships was consistent between models. All relationships with cortical volumes (paths a, b, and c) were positive. Genera mediated by GABA-related genes were all positively related to short-term memory and inversely related to *GAD* or *SLC32A1*, which themselves were inversely related to memory. In short, downregulation of GABA neurotransmission-related genes and increased cortical volume mediated the positive relationship between short-term memory and relative abundance of *Lactobacillus*, *Oscillospira*, and unclassified *Ruminococcaceae*, *Christensenellaceae*, and *Elusimicrobiaceae.* A similar phenomenon was found for the BMOS + HMO group regarding long-term memory.

In the BMOS + HMO group, mediating variables included *GABRA2*, *GRIN2B*, *SLC17A6*, *SLC17A8*, and average visit time to the novel object. Here, all mediators except for *SLC17A6* positively predicted long-term recognition memory. Each of these mediated more than one bacterial genus. Fecal *Blautia* was mediated by *GABRA2*, *GRIN2B*, and average visit time to the novel object. Fecal *Dorea* was mediated by *GABRA2* and *SLC17A8*. Fecal *Acidaminococcus* was mediated by *GABRA2* and *SLC17A6*. Here, lower relative amounts of *Blautia* and higher relative amounts of *Dorea* and *Acidaminococcus* were related to improved long-term memory.

It was unsurprising that *GABRA2*, *GRIN2B*, *SLC17A6*, and *SLC17A8* would be related to recognition memory. Each of these has been shown in some fashion to play a role in learning and memory. The expression of *GABRA2* in the frontal cortex is associated with cognitive decline [63]. Inhibition of *GRIN2B* impairs memory [64], reduces spine density [65], and it’s expression decreases with age [66]. The role of *SLC17A6* is less clear, however, *SLC17A6* is highly expressed during early development in subcortical structures [67,68], and age-impaired rats demonstrate increased levels of *SLC17A6* as compared to younger animals, with increased levels of *SLC17A6* related to cognitive deficit and lower expression of glutamatergic receptors [69]. Lastly, *SLC17A8* deletion has been shown to induce anxiety related behavior [70] and mild impairments in learning and memory [71]. The relationships of genera to these mediating variables suggests that the excitatory/inhibitory balance is an important nexus through which host bacteria can influence behavior.

These results echo those shown by Bravo et al., [26] who demonstrated that supplementation with *L. rhamnosus* reduced expression of GABA receptors in subcortical regions such as the hippocampus and amygdala simultaneous with decreases in anxiety- and depression-like behaviors. Yet, overall, there are little data linking microbial genera to glutamatergic neurotransmission in a neurotypical context, highlighting the novelty of these findings. *Blautia* has been shown to be inversely associated with sociability and positively correlated with repetitive and anxiety-like behaviors in a Spearman correlation that pooled control mice in a model of autism spectrum disorder (BTBR *T^+^ Itpr3^tf^/J* mice) [29]. In that model, a reduction in abundance of *Blautia* was found, concurrent with impairments in bile acid and tryptophan metabolism (despite correlation analysis revealing that reduced *Blautia* was related with improved sociability and anxiety outcomes). Another study investigating autism spectrum disorder (ASD) in children found reductions in *Dorea formicigenerans* and *Blautia luti* [30], but these genera were not related to serotonin, tryptophan, or cytokine-related outcomes. In a Shank3 knock-out (KO) mouse model of ASD, Shank3 KO mice demonstrated reduced hippocampal gene expression of *GABRA1*, *GABRA2*, *GABRB1* and abundance of *L. reuteri*. A positive relationship between these GABAergic genes and *L. reuteri* was found. In a follow-up, Shank3 KO mice supplemented with *L. reuteri* showed increased hippocampal and prefrontal cortex gene expression of GABAergic receptors in males and females [72]. Clinical data from a large cohort study (Flemish Gut Flora Project, n = 1054) demonstrated that microbial metabolic pathways related to GABA synthesis and glutamate degradation were increased and decreased in subjects with depression, respectively. However, pathways most related to quality of life included DOPAC synthesis, isovaleric acid synthesis, and histamine synthesis [28].

In addition to glutamatergic-related genes, we also observed multiple receptors (*GABRA5*, *5HTR2*, *CHRNA7*, *GRIA2*, *GRIN2D*) for different neurotransmitters (GABA, serotonin, acetylcholine, glutamate) that were all inversely related to short-term memory, and mediated relationships with genera in the colon such as *Prevotella*, unclassified *Prevotellaceae*, *Anaerotruncus*, and *Megasphaera* in the OF + 2′-FL group. Here, the positive relationship between colonic *Prevotella* and short-term memory was mediated by its inverse relationship to all of the aforementioned receptors. In contrast to a “convergence” upon similar mediators, here *Prevotella* diverged upon multiple different genes encoding receptors with a variety of functions, yet all bore the same inverse relationship to short-term memory.

#### 4.1.2. Myelination and Transcription Factors

Up to this point, the discussion has centered around GABA/Glu-related gene expression. However, a combination of myelin- and transcription-related genes were repeatedly found as mediators of multiple genera and long-term memory in pigs fed BMOS. Of the mediators identified (*MAG*, *MBP*, *PLP*, *NR4A2*, and *CREBBP*, all of which were inversely related to memory), *Ruminococcus* and long-term memory were mediated by all of them. There appear to be relatively little data linking myelination and gut bacteria, as compared to data linking gut bacteria to various neurotransmitter systems. Germ-free mice exhibit hyper-myelinated axons in the prefrontal cortex [73], whereas FMT from nonobese diabetic mice to C57BL/6 mice reduced prefrontal cortex myelination [22]. A human/animal trial has provided some mechanistic evidence linking the gut microbiome with myelination. Pre-term infants were categorized as “low-growth” or “high-growth” according to body weight and FMT from these infants were conducted on eight- to nine-week-old germ-free pregnant mice. Pups from germ-free mice colonized with feces from “low-growth” preterm infants demonstrated reduced NeuN, neurofilament-L, and MBP in cortical homogenates as compared to those born from dams colonized with feces from “high-growth” infants [74]. Reduction of these proteins was age dependent, as MBP was impaired at 2 weeks, but NeuN and NFL were only impaired at 4 weeks. At 4 weeks, germ-free and “low-growth” mice tended to show increased expression of glutamatergic-, GABAergic-, and ion channel-related genes, with downregulation of serotonergic and dopaminergic genes in total brain homogenate. In part, these data corroborate our own findings. We found that reduced GABAergic- and myelin-related genes, but increased expression of glutamatergic genes, was related to improved recognition memory in a diet-dependent manner. While the direction of change (whether increasing GABAergic/myelin gene expression is “good”) differs, both studies have centered around these candidates.

Even less is known regarding transcriptional regulation in the brain and host microbiome. We found *CREBBP* and *NR4A2* to mediate relationships between several genera and long-term memory but are unaware of existing data supporting this finding.

#### 4.1.3. Exploratory Behavior

With respect to long-term memory in the HMO group, total distance moved during the task was a mediator for 12 colonic and fecal genera, four of which were negatively related to long-term memory, eight of which were positively related to recognition memory (Figure 4). Otherwise, the only other mediators included *GABRD* and relative volume of the corpus callosum, which mediated the relationships between colonic *Eubacterium* and fecal unclassified *Paraprevotellaceae*, respectively. Thus, these genera likely did not directly affect recognition memory, but affected general exploratory behavior, which itself is closely related to memory. Conversely, it may be likely that the reverse is true, where highly exploratory animals were more likely to demonstrate long-term memory and have higher/lower abundance of these genera.

Several studies have shown relationships between the gut microbiome and locomotor activity (typically measured as distance moved during a behavioral task). Mice provided an oral antibiotic show slightly less locomotor activity compared to baseline [22], germ-free mice or mice humanized via FMT from schizophrenic patients show increased locomotor activity and reduced anxiety-like behavior [21,75], zebrafish given a probiotic demonstrate increased distance moved during shoaling-related behaviors [76], and pigs fed HMO demonstrated reduced distance moved during the habitation trial of the NOR task [33]. Clearly, both insults to and developmental support of the microbiome result in various effects on locomotor activity across species, which must be interpreted in context of the present study. Here, although pigs fed HMO demonstrated the least movement [33], within that group pigs with the highest locomotor activity demonstrated improved long-term recognition memory.

Average time per visit to the novel object was also a mediator between genera in the BMOS + HMO group for both short- and long-term memory. Here, visits that lasted longer tended to correlate with improved performance. The genera this variable mediated differed between short- and long-term memory and were both negatively and positively related to average visit time. Previously, we found that BMOS + HMO improved long- but not short-term memory [33].

#### 4.1.4. Brain Volume

Volume of the brain (and its subregions) was rarely a mediating a variable between genera and long-term memory (with the exception of caudate volume in pigs fed OF and relative volume of the corpus callosum in pigs fed HMO). For short-term memory, volume of the cortices, pons, and whole brain were mediators in the HMO, BMOS + HMO, and OF groups, respectively. Genera within these models included *Lactobacillus*, *Prevotella*, and *Desulfovibrio*. It was clear, however, that cortical volume (both absolute and relative) was a frequent mediator for multiple genera including *Oscillospira*, *Lactobacillus*, and unclassified genera of *Ruminococcaceae*, *Christensenellaceae*, and *Elusimicrobiaceae* in the HMO group. For each mediation, the strength of the mediating variable on memory was moderate (for every rank increase in brain volume, short-term memory increased by approximately 0.4–1 ranks). This aligns with the observation that pigs fed HMO (in the HMO or BMOS + HMO group) demonstrated increased relative volumes of the left and right cortex. Despite this finding, it is unclear whether structural volume as a mediator would persist into later life.

Carlson et al., [27] investigated the relationship between infant microbiome, cognition, and neuroimaging outcomes. Infants were grouped into three clusters based on cluster analysis of the microbiome. These groups tended to cluster around relative abundance of *Faecalibacterium*, *Bacteroides*, and an unclassified genus of *Ruminoccocaceae*. At 2 years of age, scores on the Mullen Scales of Early Learning were different between clusters and differed by receptive language and expressive language, with the group clusters around *Bacteroides* performing the best. Some associations were found between alpha diversity and regional gray matter volume, but most regions measured were not different between clustered groups. The authors suggest that the gut microbiome may have minimal effects on regional brain volume by 1–2 years of age. When comparing germ-free to specific pathogen free (SPF) mice, SPF mice tend to show higher regional volume, fractional anisotropy, macromolecular proton fraction (marker of myelination), and myelin staining in numerous regions [77]. Yet, these differences largely disappeared between 4 and 12 weeks, while behavioral differences persisted, further confirming the hypothesis by Carlson et al., [27] that structure of the brain is minimally impacted by the gut microbiome. Furthermore, we observed no evidence of spectroscopic or fractional anisotropy measures as mediating variables. Based on the number of mediating variables that included genes, it would appear that molecular function better describes the relationship between cognition and gut bacteria than does structure or metabolic composition of the brain.

### 4.2. Potential Mechanisms of Action of Prebiotics

The objective of this analysis was to identify potential pathways between gut bacteria and cognition and likely mechanistic paths between the two. Ultimately, the goal was to identify a mechanistic path that might explain the differential effects that various types of oligosaccharides have on their host. However, while new data were generated, as usual in most scientific research, more questions were raised than answered. The clearest finding however was that the action of microbial genera on memory was frequently associated with GABAergic and glutamatergic genes.

Several mechanisms have been put forth linking the gut and brain. These include vagal activity, the neuroimmune axis, and circulation of microbial metabolites [78]. The present data do not suggest any of these routes are inaccurate. However, it may be that glutamatergic activity is modulated by a combination of these factors. It is likely that numerous mediators may be identified between the gut and brain, however the question remains as to which are the most likely candidates. Our findings support the hypothesis that the human milk oligosaccharides used in this study act via alteration of hippocampal expression of GABAergic and glutamatergic genes, most likely in turn impacting long-term potentiation to improve memory, and long-term potentiation itself is what may be mediated by another pathway [17,26].

Beyond shedding light on potential biomarkers, the present analysis highlights the importance of including groups in statistical modelling (whether as a sub-group analysis or as categorical predictors in the model). Given the presence of mediators in some groups and not others, it is likely that they would not have been identified if correlation or regression analyses were conducted irrespective of treatment level. Furthermore, the identification of multiple suppressor variables suggests that simple correlation matrices might result in a significant number of false negatives, or at least underestimate the true relationship between predictor and dependent variables.

### 4.3. Limitations

Important limitations of the current analysis should be taken into consideration. First, all data were ranked prior to analysis. As described previously, this sacrifices reproducibility across studies but provides the benefit of attenuating non-normality, skew, and the weight of outliers. Second, a sub-group analysis was performed, rather than performing a moderated mediation analysis. Combined with the lack of equal sample size across models a sub-group analysis may lead to the inaccurate assessment that a lack of mediators in one group versus another represents a true biological difference. The data should be viewed in context of these limitations, with the understanding that other methods of modelling the data may reveal different insights. Lastly, 16S sequencing, mRNA copy number, and structural MRI outcomes do not necessarily equate to relevant biological activity. Similarly, mRNA copy number and MRI outcomes do not directly equate to biological activity However, these data present compelling evidence that may be used to justify further, more rigorous, and well-controlled examinations into the gut–brain axis.

## 5. Conclusions

We conducted a sub-group mediation analysis between gut bacteria, brain outcomes, and cognition. Gene expression of GABA and glutamate-related neurotransmission frequently mediated the relationship between gut bacteria and both short- and long-term memory. Additional mediators included myelination-related genes, transcription factors, brain volume, and exploratory behavior. Importantly, mediating variables were not equally present in all dietary groups. These data provide a step in the understanding of pathways mediating the association between dietary intake of prebiotics in early life and memory function. In particular, these data provide further support for a key role of hippocampal GABAergic and glutamatergic genes in the link between gut and brain development. These data should help identify potential biomarkers and mechanistic pathways for future research.

## Figures and Tables

**Figure 1 microorganisms-09-00846-f001:**
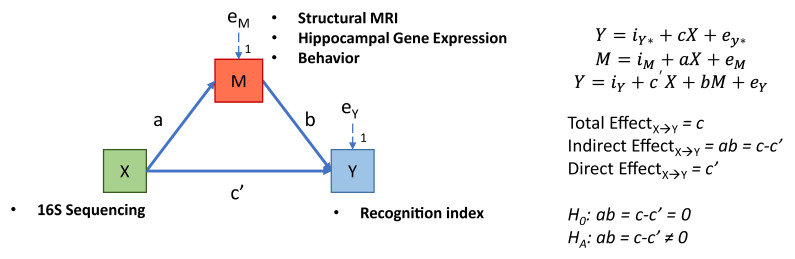
Schematic depicting the mediation models and regression paths. The objective of the mediation analysis was to determine if the indirect effect (paths a*b, which is equal to paths c-c’) was different from zero, suggesting the mediating variable altered the strength of the relationship between X and Y. Three models were used to assess the mediation: X regressed on Y, X regressed on M, and both X and M regressed on Y. The letters a, b, c, and c’ refer to estimates the beta coefficients for each respective model, with e representing the error term.

**Figure 2 microorganisms-09-00846-f002:**
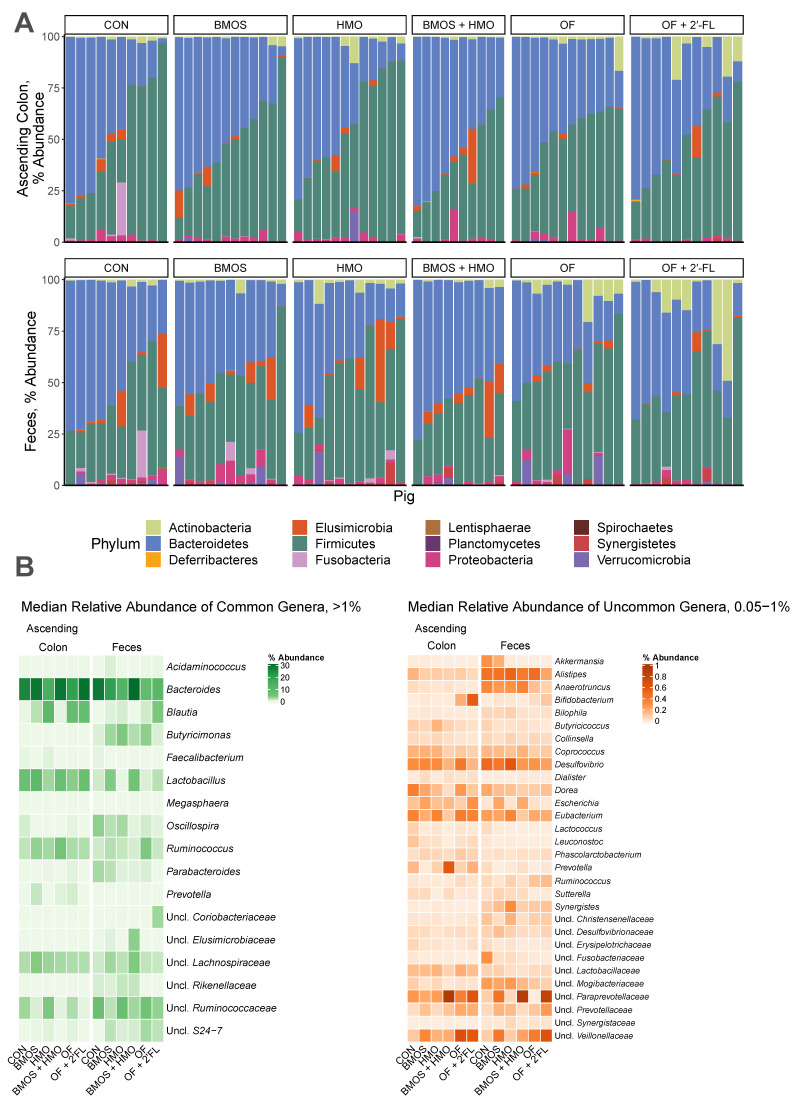
(**A**) Relative abundance of taxa at the phylum level in the ascending colon and feces. Each bar represents an individual pig. (**B**) Relative abundance of common and uncommon taxa at the genera level. Groups with a median abundance less than 0.05% for any given taxa are not shown. Abbreviations: Uncl., unclassified; CON, control group; HMO, pigs fed human milk oligosaccharides; BMOS, pigs fed bovine milk oligosaccharides; BMOS + HMO, pigs fed both human and bovine milk oligosaccharides; OF, pigs feds oligofructose; OF + 2′-FL, pigs fed oligofructose and 2′-fucosyllactose.

**Figure 3 microorganisms-09-00846-f003:**
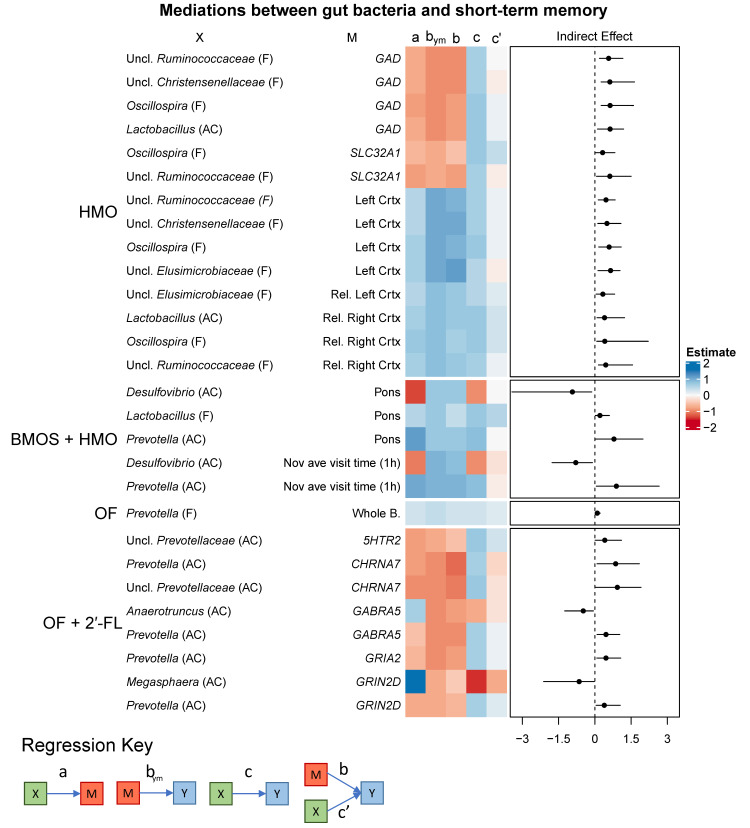
Heatmaps demonstrating the relationships between predictor (X), mediating (M), and predicted variables (Y, short-term recognition memory). Only mediations whose confidence intervals of the indirect effect do not include zero are shown. The right-most box represents the estimates of the indirect effect (c-c’) bounded by its lower and upper 95% confidence intervals. Paths a, b_ym_, b, c, and c’ are the regression coefficients for X → M, M → Y, X → Y, and X + M → Y, respectively. Abbreviations: Uncl., unclassified; CON, control group; HMO, pigs fed human milk oligosaccharides; BMOS, pigs fed bovine milk oligosaccharides; BMOS + HMO, pigs fed both human and bovine milk oligosaccharides; OF, pigs feds oligofructose; OF + 2′-FL, pigs fed oligofructose and 2′-fucosyllactose.

**Figure 4 microorganisms-09-00846-f004:**
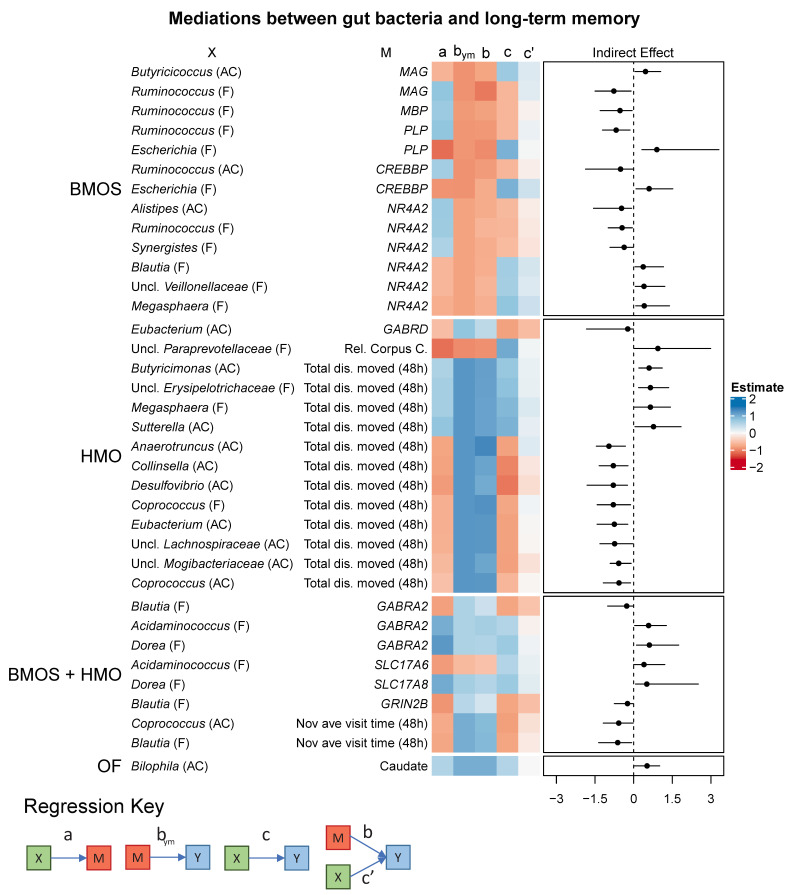
Heatmaps demonstrating the relationships between predictor (X), mediating (M), and predicted variables (Y, long-term recognition memory). Only mediations whose confidence intervals of the indirect effect do not include zero are shown. The right-most box represents the estimates of the indirect effect (c-c’) bounded by its lower and upper 95% confidence intervals. Paths a, b_ym_, b, c, and c’ are the regression coefficients for X → M, M → Y, X → Y, and X + M → Y, respectively. Abbreviations: Uncl., unclassified; CON, control group; HMO, pigs fed human milk oligosaccharides; BMOS, pigs fed bovine milk oligosaccharides; BMOS + HMO, pigs fed both human and bovine milk oligosaccharides; OF, pigs feds oligofructose; OF + 2′-FL, pigs fed oligofructose and 2′-fucosyllactose.

**Table 1 microorganisms-09-00846-t001:** Carbohydrate content of the diets ^1^.

	Bovine Milk Oligosaccharides ^2^		2′Fucosyllactose ^3^		Lacto-N-Neotetraose ^3^		Oligofructose ^4^		Additional Lactose		Total OS		Total Lactose		Total Carbohydrate	
Diet	Per kg	Per L	Per kg	Per L	Per kg	Per L	Per kg	Per L	Per kg	Per L	Per kg	Per L	Per kg	Per L	Per kg	Per L
Formulated																
CON	0	0	0	0	0	0	0	0	69.30	13.86	0	0	412.79	82.56	412.79	82.56
BMOS	61.98	12.40	0	0	0	0	0	0	61.98	1.47	61.98	12.40	350.82	70.16	412.79	82.56
HMO	0	0	4.92	0.98	2.41	0.48	0	0	7.33	12.40	7.33	1.47	405.47	81.09	412.79	82.56
BMOS + HMO	61.98	12.40	4.92	0.98	2.41	0.48	0	0	0	0	69.30	13.86	343.49	68.70	412.79	82.56
OF	0	0	0	0	0	0	26.08	5.22	61.98	8.64	26.08	5.22	386.71	77.34	412.79	82.56
OF + 2′-FL	0	0	4.92	0.98	0	0	26.08	5.22	38.30	7.66	31.00	6.20	381.79	76.36	412.79	82.56
Analyzed																
CON	0	0	0	0	0	0	0	0	NQ	NQ	0	0	NQ	NQ	NQ	NQ
BMOS	28.95	5.79	0	0	0	0	0	0	NQ	NQ	28.95	5.79	NQ	NQ	NQ	NQ
HMO	0	0	4.05	0.81	2.10	0.42	0	0	NQ	NQ	6.15	1.23	NQ	NQ	NQ	NQ
BMOS + HMO	28.75	5.75	5.00	1.00	2.65	0.53	0	0	NQ	NQ	36.40	7.28	NQ	NQ	NQ	NQ
OF	0	0	0	0	0	0	18.10	3.62	NQ	NQ	18.10	3.62	NQ	NQ	NQ	NQ
OF + 2′-FL	0	0	5.60	1.12	0	0	17.05	3.41	NQ	NQ	22.65	4.53	NQ	NQ	NQ	NQ

^1^ Abbreviations: OS, oligosaccharide; CON, control group; HMO, pigs fed human milk oligosaccharides; BMOS; pigs fed bovine milk oligosaccharides, BMOS + HMO, pigs fed both human and bovine milk oligosaccharides; NQ, not quantified. ^2^ Nestlé Product & Technology center, Konolfingen, Switzerland. ^3^ Glycom, Hørsholm, Denmark. ^4^ Orafti^®^ P95; Beneo-Orafti, Tienen, Belgium.

## Data Availability

Data may be provided upon request.

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
