# Peer review of "A Mediation Analysis to Identify Links between Gut Bacteria and Memory in Context of Human Milk Oligosaccharides"

_microorganisms, 2021, doi:10.3390/microorganisms9040846_

Round 1

Reviewer 1 Report

The results of the research is interesting and seems to have nice potentials on pigs. However, the results seems not enough to demonstrate what the authors want to suggest. The revise of some parts are needed

  1. Did you consider administrating Colostridium perfingens antitoxin C+D when you analyze the microbiome results? Because it may affect to the microbiota, or are there any reference papers that antitoxin do not affects the microbiome?
  2. Pigs are sometimes eat their own fecal, did you monitor or consider its characteristics?
  3. Need to demonstrate abbreviation of OS in line 109.
  4. The additional demonstration of the groupings is needed, or may show additional figure as image.
  5. What is the purpose of setting two OF groups?
  6. Group demonstration in 2.2 paragraph is not understandable, rewrite clearly.
  7. The title of the label under each figures need to be re-written in short and clear way. (ex. Remove ‘Here,’ after Heatmaps demonstrating ~~~ sentence)
  8. All of the figure seems blur, the resolution needs to be improved
  9. Are there any additional brain MRI images or image analyzed data to show?
  10. Any other data or results such as body weight, feed intake, and so on?

Author Response

The results of the research is interesting and seems to have nice potentials on pigs. However, the results seems not enough to demonstrate what the authors want to suggest. The revise of some parts are needed

  1. Did you consider administrating Colostridium perfingens antitoxin C+D when you analyze the microbiome results? Because it may affect to the microbiota, or are there any reference papers that antitoxin do not affects the microbiome?
    1. C. perfringens antitoxin likely does impact the microbiome, but this is a necessary and standard procedure to ensure the health of the animals. Newborn pigs are especially susceptible to infection from this pathogen [1]. We will make this known in the manuscript.
  2. Pigs are sometimes eat their own fecal, did you monitor or consider its characteristics?
    1. Pigs are indeed known to practice coprophagy, especially in a group setting. This was not directly observed, but subjective reports suggest in an artificially reared environment the incidence of this happening is low. It is likely not clear to readers, but the flooring used is slotted and allows fecal and urinary waste to drop below the cage into a plumbing system.
  3. Need to demonstrate abbreviation of OS in line 109.
    1. Amended.
  4. The additional demonstration of the groupings is needed, or may show additional figure as image.
    1. We have added a table on dietary treatments.
  5. What is the purpose of setting two OF groups?
    1. We have added a table on dietary treatments. To directly answer your question, these two groups contained either oligofructose alone, or oligofructose and 2’fucosyllactose. This was the subject of previous research cited [2], where we investigated if additional 2’fucosyllactose improves cognitive outcomes when oligofructose was already a part of the diet.
  6. Group demonstration in 2.2 paragraph is not understandable, rewrite clearly.
    1. We have added a table on dietary treatments.
  7. The title of the label under each figures need to be re-written in short and clear way. (ex. Remove ‘Here,’ after Heatmaps demonstrating ~~~ sentence)
    1. The text under figure 1 and 4 were errors, they have been amended. Others have been made more concise.
  8. All of the figure seems blur, the resolution needs to be improved.
    1. High resolution images will be used for the accepted manuscript.
  9. Are there any additional brain MRI images or image analyzed data to show?
    1. Yes, however these data are presented in the cited papers [2,3]. There was simply too much data to incorporate those images here as well.
  10. Any other data or results such as body weight, feed intake, and so on?
    1. These are detailed in the cited papers [2,3], as well as in a companion paper submitted simultaneously to this one [4].

  1. Niilo, L. Clostridium perfringens Type C Enterotoxemia. Can. Vet. J. = La Rev. Vet. Can. 1988, 29, 658–64.
  2. Fleming, S.A.; Mudd, A.T.; Hauser, J.; Yan, J.; Metairon, S.; Steiner, P.; Donovan, S.M.; Dilger, R.N. Dietary Oligofructose Alone or in Combination with 2′-Fucosyllactose Differentially Improves Recognition Memory and Hippocampal mRNA Expression. Nutrients 2020, 12, 2131.
  3. Fleming, S.A.; Mudd, A.T.; Hauser, J.; Yan, J.; Metairon, S.; Steiner, P.; Donovan, S.M.; Dilger, R.N. Human and Bovine Milk Oligosaccharides Elicit Improved Recognition Memory Concurrent With Alterations in Regional Brain Volumes and Hippocampal mRNA Expression. Front. Neurosci. 2020, 14, 1–14.
  4. Wang, M.; Monaco, M.H.; Hauser, J.; Yan, J.; Dilger, R.N.; Donovan, S.M. Bovine Milk Oligosaccharides and Human Milk Oligosaccharides Modulate the Gut Microbiota Composition and Volatile Fatty Acid Concentrations in a Neonatal Model. Microorganisms 2021.

Reviewer 2 Report

The manuscript by Fleming et al. entitled “A mediation analysis to identify links between gut bacteria and memory in context of human milk oligosaccharides”, submitted for my peer-review, explores the inscrutable area of mediators linking gut microbiome with cognition. This topic is of great interest to scientific community as it may help in deeper understanding of the gut-brain axis and may speed the research in this area. The study does not test any specific scientific hypothesis; conversely, it generates hypotheses that warrant further investigation.

Although I have a limited expertise in some methods used, the study appears to me methodologically adequate and robust. The conclusions are supported by the obtained results and presented in a balanced way, appropriate for this kind of explorative research. Several minor issues require to be addressed before the manuscript is suitable for publication. Particularly, 1) the effect of oligosaccharide type consumed should be balanced as it is not specifically tested and the analysis finds no much difference and 2) the study should be clearly stated “a secondary analysis” of datasets that previously published papers were based on – if this is the case.

Below, I present all my comments. They are presented in the order of as they appear in the manuscript, but not according to their significance.

Lines 25-26

I recommend balancing or even removing the message that “…genera were related to memory in a diet-dependent manner”. The Authors do not specifically test the difference between the diets and Figure 2 suggests similarity of relative abundance of bacteria between diets.

Line 38

I am not a native English speaker. But should it be rather something like “Human milk contains a variety of quality and concentration of oligosaccharides…”?

Line 78

What is the relationship between the present study and the previous Authors’ papers cited as [32, 33]? Is the present study fully separate from the recent studies of the Authors (that were cited in-text as [32, 33])? Or is it a reanalysis of the existing data as expressed in the Abstract (lines 22-23)? This should be clearly disclosed in the Methods section and the study called “secondary analysis” if this is the case. Additionally, suitable limitations of such “secondary analysis” approach should be discussed.

Line 84

How were the animals allocated to the groups? Randomly? How was it performed? Was all the sample of 72 pigs randomized at the same moment? Or were the cohorts built separately for the purpose of previously published papers? If this is the case, were the cohorts comparable? Also, shortly explain what was the difference between the cohorts as early as in this line (eg. “…across six independent cohorts (n=12 per cohort) differing only by dietary treatments.”).

Lines 96-97

Why were the pigs administered Cl. perfingens antitoxin? To protect from food poisoning? Is it a standard procedure? How does it affect intestinal microbial abundance?

Lines 104, 109-113

Explain the abbreviations: OS, BMOS, HMO as early as they first appear in-text.

Lines 108

What does it mean “were analyzed to contain”? Did the Researchers conducted independent analysis of the content of diets? What does it mean “contain approximately 0 g/L”? A few milligrams, micrograms, nanograms…? What is the level of negligible/insignificant amount of OS in the diet?

Lines 109-115

Please provide the details of six diets from the new bullet points each (for better legibility).

Line 118

Does “…same amount of total carbohydrate” refer to digestible carbohydrates or the total, as it is written?

Line 139

Please underline that “delay of 1- or 48-h” represents “short- and long-delay”, respectively, as referred to later on in the manuscript.

Line 170

What were the housekeeping genes included?

Line 194

“(51)” is it a citation or number?

Lines 199-206

There is no need to repeat the conditions of variable specification in the footnote. The Authors may refer the Readers to the text. The footnote should rather include the description of the analysis plan depicted with mathematical formulas. In particular what do the symbols of formulas mean and how are they relevant to the aim/hypotheses of the study. Where is “c” presented in the graph? Does “e” represent error? Etc.

Line 200

Something appears wrong in 2nd line of the footnote of the Figure. Figure 2 is cited? Or the Authors mean two SDs?

Lines 215-217

Please number consecutive variables in Table S1. It would help search the list. Also, provide the total number of analyzed variables in Data analysis subsection in the main text. What were the sources of missingness? What was the fraction of values within database missing? How were “non-quantified variables” coded in the analyses?

Lines 243, 262-263

What is “confidence intervals of the indirect effect > |0|”? Should not be rather “|confidence intervals of the indirect effect| > 0”? Or just descriptively “CIs do not include zero”?

Figure 2

Do not refer in the footnote to panels (top or bottom) but A or B as indicated in the Figure. What do the vertical stripes in part A of the Figure mean (there are about 10-11 such strips for each cohort and each sample type of feces/colon)? Are they individual data (separate for each animal)? If this is the case, why were there no 12 strips? Only two pigs were reported to be excluded from the analysis (lines 103-105). But line 568 also suggests more dropouts. Were there any statistical analyses performed to determine whether the type of diet significantly affects the abundance of bacteria?

Figure 3

Please repeat here what are the predictor and tested mediators (line 261). In Fig 2 regression coefficient X-to-Y is called c’, whereas in Fig 3 it is just c (similarly b and bym appear confused). What does bym represent? In lines 264-265 five paths seem to be related to four regressions. The names of genes and MRI data should be explained in the footnote. Why do total effects (c) differ between the type of mediator? Should they link predictors and predicted values in total (with no consideration of mediators)? It would be valuable if the total effects (X-to-Y) were reported for all the six cohorts regardless the effect of mediation (even if it was reported in previous papers of the Authors). Similar issues should be addressed to Fig 4.

Lines 248-249

Why is microbial relative abundance (the predictor) based on fecal, but not ascending colon samples (apart from Lactobacillus)? Why is Lactobacillus an exemption? Did the Authors perform sensitivity analysis with ascending colon samples for all the genera? Are the analyses in line with the ones including fecal samples?

Lines 274-275

Is “binding protein” an redundant repetition? Should it be “cyclic AMP response element binding protein”?

Line 308

Square brackets for citation.

Line 309

“Were” rather than “was”.

Line 325

This is difficult to say that the mediating effects differed between the groups as the relationships within groups were not compared to each other with any statistical test. For example no mediating effect was found for CON and several such effects were found in the other groups. However, this is possible that p-values for mediating effects in CON was just above the significance threshold, whereas in other groups p-values could be just below. Moreover, the groups could present different statistical power to detect the effects as discussed later on (lines 359-361). I would advice to soften this expression to something like “appeared to differ”.

Lines 328-329

Did Authors try to log-transform or apply Box-Cox transformation on their variables to address this problem? This could save continuity of the variables.

Lines 365-366

The problem of no false discovery rate correction should be addressed also by providing p-values for each mediation (indirect) effect detected in Fig 3, 4 and S1. Readers interested in looking for more “assured” results could assess how much the effect is unlikely a chance by inspecting p-values.

Line 379-380, 392-393

Please remind here the exact function of analyzed genes. Moreover, how the Authors explain the negative link of GABA genes with memory, whereas largely positive link of glutamate genes? Just well-known GABA-glutamate opposition?

Line 419

Capitalize the first letter of “spearman”

Lines 448-450

Were the directions of mediation (positive or negative) consistent for these genes?

Line 454

FMT – fecal microbial transplant? The abbreviation should be explained.

Limitations

Another limitation is that mRNA gene expression profile was tested, but not the biological activity of the gene products. Changes in mRNA expression may arise as compensatory reactions to the modulation in biological activity of receptors/enzymes/carriers/etc. and thus may represent the opposite direction of the effect. Please include this limitation.

Lines 587-588

Please complete the Supplementary Materials statement.

Author Response

The manuscript by Fleming et al. entitled “A mediation analysis to identify links between gut bacteria and memory in context of human milk oligosaccharides”, submitted for my peer-review, explores the inscrutable area of mediators linking gut microbiome with cognition. This topic is of great interest to scientific community as it may help in deeper understanding of the gut-brain axis and may speed the research in this area. The study does not test any specific scientific hypothesis; conversely, it generates hypotheses that warrant further investigation.

Although I have a limited expertise in some methods used, the study appears to me methodologically adequate and robust. The conclusions are supported by the obtained results and presented in a balanced way, appropriate for this kind of explorative research. Several minor issues require to be addressed before the manuscript is suitable for publication. Particularly, 1) the effect of oligosaccharide type consumed should be balanced as it is not specifically tested and the analysis finds no much difference and 2) the study should be clearly stated “a secondary analysis” of datasets that previously published papers were based on – if this is the case.

Below, I present all my comments. They are presented in the order of as they appear in the manuscript, but not according to their significance.

To specifically answer the question of the unbalanced design in concentration, we purposely used concentrations that replicated those used in previous clinical trials, which we describe in section 2.2. We note this is indeed a limitation of the design, but the trial cannot at this point be conducted again.

Lines 25-26

I recommend balancing or even removing the message that “…genera were related to memory in a diet-dependent manner”. The Authors do not specifically test the difference between the diets and Figure 2 suggests similarity of relative abundance of bacteria between diets.

We agree that the median abundance of genera appears similar between diets, and we did not do a direct comparison between them. Those are present in a companion paper submitted simultaneously to this one [1] and we have added a reference to these data in the introduction and results. In this manuscript, the focus was not to investigate mean differences in genera between groups in a univariate manner, but specifically to investigate how they relate (regress to / correlate with) memory. We agree that a moderated mediation could have been performed to directly assess if the relationships themselves differed between diets but describe why this was not possible in the beginning of the discussion. Despite these limitations, we believe the data are quite clear in that the relationships identified between genera and memory do indeed vary between diets, and even between short- and long-term memory. We will revise to state “Relationships between genera and memory appeared to differ between diets”, which has a better tone but states the same intent.

Line 38

I am not a native English speaker. But should it be rather something like “Human milk contains a variety of quality and concentration of oligosaccharides…”?

We’ve revised the sentence to improve readability.

 Line 78

What is the relationship between the present study and the previous Authors’ papers cited as [32, 33]? Is the present study fully separate from the recent studies of the Authors (that were cited in-text as [32, 33])? Or is it a reanalysis of the existing data as expressed in the Abstract (lines 22-23)? This should be clearly disclosed in the Methods section and the study called “secondary analysis” if this is the case. Additionally, suitable limitations of such “secondary analysis” approach should be discussed.

Based on your comments further on in the manuscript, it is clear that the limitations were appreciated. The analyses are exactly as stated, though we added language to highlight this is a secondary analysis in the introduction and results.

 Line 84

How were the animals allocated to the groups? Randomly? How was it performed? Was all the sample of 72 pigs randomized at the same moment? Or were the cohorts built separately for the purpose of previously published papers? If this is the case, were the cohorts comparable? Also, shortly explain what was the difference between the cohorts as early as in this line (eg. “…across six independent cohorts (n=12 per cohort) differing only by dietary treatments.”).

We more clearly stated that randomization was performed to achieve equal litter representation and body weight within each cohort and between each dietary group, respectively. We also noted that cohorts were reared at different times. As a note to the reviewer, but not in the manuscript: There are frequently cohort differences in most animal and even human studies. However, this is why representation was balanced equally across diets, such that the effect of cohort was random but also equally present across each diet.

Lines 96-97

Why were the pigs administered Cl. perfingens antitoxin? To protect from food poisoning? Is it a standard procedure? How does it affect intestinal microbial abundance?

  1. perfringens antitoxin likely does impact the microbiome, but this is a necessary and standard procedure to ensure the health of the animals [2]. Newborn pigs are especially susceptible to infection from this pathogen. We will make this known in the manuscript.

Lines 104, 109-113

Explain the abbreviations: OS, BMOS, HMO as early as they first appear in-text.

Amended.

Lines 108

What does it mean “were analyzed to contain”? Did the Researchers conducted independent analysis of the content of diets? What does it mean “contain approximately 0 g/L”? A few milligrams, micrograms, nanograms…? What is the level of negligible/insignificant amount of OS in the diet?

Yes. It is common in nutritional studies for the actual (or analyzed) diets to contain different concentrations of supplemental ingredients than what was formulated. We have added a table on the dietary composition to answer these questions.

Lines 109-115

Please provide the details of six diets from the new bullet points each (for better legibility).

A table has been provided to clarify the diets.

Line 118

Does “…same amount of total carbohydrate” refer to digestible carbohydrates or the total, as it is written?

See above answer, a table has been provided. It refers to total carbohydrate, not just digestible.

Line 139

Please underline that “delay of 1- or 48-h” represents “short- and long-delay”, respectively, as referred to later on in the manuscript.

Amended.

Line 170

What were the housekeeping genes included?

Added. Ribosomal Protein L19,RPL-19; Ribosomal Protein S18, RPS18; Actin β, ACTB; Tubulin β Class 1, TUBB; Glyceraldehyde-3-Phosphate Dehydrogenase, GAPDH; β-glucoronidase, GUS.

Line 194

“(51)” is it a citation or number?

Amended, a citation.

Lines 199-206

There is no need to repeat the conditions of variable specification in the footnote. The Authors may refer the Readers to the text. The footnote should rather include the description of the analysis plan depicted with mathematical formulas. In particular what do the symbols of formulas mean and how are they relevant to the aim/hypotheses of the study. Where is “c” presented in the graph? Does “e” represent error? Etc.

The text here was an error, it has been amended.

Line 200

Something appears wrong in 2nd line of the footnote of the Figure. Figure 2 is cited? Or the Authors mean two SDs?

The text here was an error, it has been amended.

Lines 215-217

Please number consecutive variables in Table S1. It would help search the list. Also, provide the total number of analyzed variables in Data analysis subsection in the main text. What were the sources of missingness? What was the fraction of values within database missing? How were “non-quantified variables” coded in the analyses?

All of the above have been added including more supplemental tables to answer these questions.

Lines 243, 262-263

What is “confidence intervals of the indirect effect > |0|”? Should not be rather “|confidence intervals of the indirect effect| > 0”? Or just descriptively “CIs do not include zero”?

Good catch on the absolute value and we have simplified this sentence.

Figure 2

Do not refer in the footnote to panels (top or bottom) but A or B as indicated in the Figure. What do the vertical stripes in part A of the Figure mean (there are about 10-11 such strips for each cohort and each sample type of feces/colon)? Are they individual data (separate for each animal)? If this is the case, why were there no 12 strips? Only two pigs were reported to be excluded from the analysis (lines 103-105). But line 568 also suggests more dropouts. Were there any statistical analyses performed to determine whether the type of diet significantly affects the abundance of bacteria?

We will amend the caption to make this data more interpretable. Each bar represents an individual pig. See above answer to your second comment for why statistical analyses on between-group differences weren’t performed.

Figure 3

Please repeat here what are the predictor and tested mediators (line 261). In Fig 2 regression coefficient X-to-Y is called c’, whereas in Fig 3 it is just c (similarly b and bym appear confused). What does bym represent? In lines 264-265 five paths seem to be related to four regressions. The names of genes and MRI data should be explained in the footnote. Why do total effects (c) differ between the type of mediator? Should they link predictors and predicted values in total (with no consideration of mediators)? It would be valuable if the total effects (X-to-Y) were reported for all the six cohorts regardless the effect of mediation (even if it was reported in previous papers of the Authors). Similar issues should be addressed to Fig 4.

The predicted variables and mediators are directly shown and labelled in the figure 3 and 4. For your other points, these are all excellent questions, highlighting the difficulty in describing such an analysis. First, it is important to note that the estimates for the regression coefficients depend on the entire regression model. Thus, the relationship between X and Y is different when the model also includes X and M regressed on Y. We added the regression key to demonstrate the estimates (a, b, bym, c, c’) of the beta coefficients for each predictor differ based on other predictors present. Yes, there are indeed five paths but four models, because one of the models includes two predictors (X + M -> Y). There are really only three models of interest, which are described in Figure 1. The regression between M and Y (bym is the relationship between M and Y, as noted in the regression key) is irrelevant for a mediation analysis, but we anticipate readers being curious about that specific relationship. The mediator is not a part of the total effect (c), because as shown in the regression key c is the relationship between just X and Y.

Regarding showing all total effects, we agree in theory. But there is such an overwhelming amount of data that in practice it becomes impossible to visualize them succinctly in a manuscript and for readers to interpret. To be clear, no mediations were reported in previous papers by the authors.

There is an indeed an error in figure 1, where the hypotheses should note that ab = c-c’. This will be fixed.

Lines 248-249

Why is microbial relative abundance (the predictor) based on fecal, but not ascending colon samples (apart from Lactobacillus)? Why is Lactobacillus an exemption? Did the Authors perform sensitivity analysis with ascending colon samples for all the genera? Are the analyses in line with the ones including fecal samples?

Predictors did include both fecal and ascending colon contents. Why we only found a relationship between fecal genera and short-term memory in the HMO group is unclear. However, discovering such findings were indeed the point of this manuscript! If by sensitivity analysis you mean an assessment of the effect size, no. However, given these relationships are spearman ranked, all mediations are represented with the same units (ranked variables), and thus relative effect sizes can be intuited by viewing the estimates (which are estimates of ranks) and their confidence intervals. Section 4.14 describes how the reader can interpret the rank estimates.

Lines 274-275

Is “binding protein” an redundant repetition? Should it be “cyclic AMP response element binding protein”?

Unfortunately not! CREBBP is the CREB-binding protein. Which when spelled out creates the awkward sentence of “cyclic AMP response element binding protein (CREB) binding protein.

Line 308

Square brackets for citation.

Amended.

 Line 309

“Were” rather than “was”.

 Amended.

Line 325

This is difficult to say that the mediating effects differed between the groups as the relationships within groups were not compared to each other with any statistical test. For example no mediating effect was found for CON and several such effects were found in the other groups. However, this is possible that p-values for mediating effects in CON was just above the significance threshold, whereas in other groups p-values could be just below. Moreover, the groups could present different statistical power to detect the effects as discussed later on (lines 359-361). I would advice to soften this expression to something like “appeared to differ”.

Language softened.

 Lines 328-329

Did Authors try to log-transform or apply Box-Cox transformation on their variables to address this problem? This could save continuity of the variables.

We have used such transformations before in previous work but did not here. In some scenarios log-transformation still fails to reduce heavy skew and the influence of outliers. Rather than try and exhaustively use various transformations, we felt it best to use to the spearman rank transformation across all variables, for reasons explained in this section. As a further note here, by using spearman rank across all variables, it allows for a simple comparison between models because all variables contain the same unit (ranks).

 Lines 365-366

The problem of no false discovery rate correction should be addressed also by providing p-values for each mediation (indirect) effect detected in Fig 3, 4 and S1. Readers interested in looking for more “assured” results could assess how much the effect is unlikely a chance by inspecting p-values.

We understand the desire to do so but will note that P-values are almost never referred to, except to note that for those mediations where CI of the indirect effect were not zero, the P-value was also less than 0.05. In fact, in the next revision we will remove the use of the word “significant” on lines 243 and any reference to P-values. We strongly desire to avoid relying on the use of P-values at all, as suggested by the American Statistical Association. The over-reliance on hypothesis testing and the use of an arbitrary p-value (or q-value in the case of FDR) threshold to determine what is “significant” conflates biological and statistical significance. This linked paper provides an excellent review of why over-reliance on P-values can be dangerous. Others have noted the use of confidence intervals and/or effect sizes are appropriate alternatives to P-values. Here, we chose to minimize the use of words like hypothesis testing, P-value, and significance, and opted to use confidence intervals as the primary way to describe the results. Given an FDR would not affect the estimates or confidence intervals, it becomes a moot point given we have not reported P-values.

Instead of using p-values to assess how “strong” the results are, the reader may assess the magnitude of the indirect effect and the span of the confidence interval to intuit the “sureness” or level of “chance”.

Line 379-380, 392-393

Please remind here the exact function of analyzed genes. Moreover, how the Authors explain the negative link of GABA genes with memory, whereas largely positive link of glutamate genes? Just well-known GABA-glutamate opposition?

We won’t go into detail in the manuscript, for the sake of brevity. The answer to your question (for lines 392-393) is provided a few lines later. Those genes were mostly related to cognitive impairment. Re: lines 379-380: GAD is glutamate decarboxylase, which produces GABA from L-glutamic acid. SLC32A1 is a vesicular GABA transporter. We note on lines 387 that these are inversely related to memory (less of them = better memory).

Yes, there is the general inhibition/stimulation role of GABA and Glutamate, with inhibition of glutamate often leading to impaired memory. That certainly oversimplifies the role of many receptors, not to mention the function of subunits of those receptors.

Line 419

Capitalize the first letter of “spearman”

Amended.

Lines 448-450

Were the directions of mediation (positive or negative) consistent for these genes?

Negative, this has been added.

 Line 454

FMT – fecal microbial transplant? The abbreviation should be explained.

Amended.

Limitations

Another limitation is that mRNA gene expression profile was tested, but not the biological activity of the gene products. Changes in mRNA expression may arise as compensatory reactions to the modulation in biological activity of receptors/enzymes/carriers/etc. and thus may represent the opposite direction of the effect. Please include this limitation.

This limitation extends to both MRI outcomes and 16S sequencing data, so we noted those in addition to mRNA as limitations.

Lines 587-588

Please complete the Supplementary Materials statement.

Amended.

  1. Wang, M.; Monaco, M.H.; Hauser, J.; Yan, J.; Dilger, R.N.; Donovan, S.M. Bovine Milk Oligosaccharides and Human Milk Oligosaccharides Modulate the Gut Microbiota Composition and Volatile Fatty Acid Concentrations in a Neonatal Model. Microorganisms 2021.
  2. Niilo, L. Clostridium perfringens Type C Enterotoxemia. Can. Vet. J. = La Rev. Vet. Can. 1988, 29, 658–64.

Round 2

Reviewer 1 Report

Every query has been successfully responded. It seems ok to be published.